# TRANSFERRABLE END-TO-END LEARNING FOR PROTEIN INTERFACE PREDICTION

## ABSTRACT

While there has been an explosion in the number of experimentally determined, atomically detailed structures of proteins, how to represent these structures in a machine learning context remains an open research question. In this work we demonstrate that representations learned from raw atomic coordinates can outperform hand-engineered structural features while displaying a much higher degree of transferrability. To do so, we focus on a central problem in biology: predicting how proteins interact with one another—that is, which surfaces of one protein bind to which surfaces of another protein. We present Siamese Atomic Surfacelet Network (SASNet), the first end-to-end learning method for protein interface prediction. Despite using only spatial coordinates and identities of atoms as inputs, SASNet outperforms state-of-the-art methods that rely on hand-engineered, high-level features. These results are particularly striking because we train the method entirely on a significantly biased data set that does not account for the fact that proteins deform when binding to one another. Demonstrating the first successful application of transfer learning to atomic-level data, our network maintains high performance, without retraining, when tested on real cases in which proteins do deform.

## 1 INTRODUCTION

Proteins are large molecules that carry out almost every function in the cell. Their function depends critically on their ability to bind to one another in specific ways, forming larger machines known as protein complexes. In this work we tackle the problem of protein interface prediction: given the separate structures of two proteins, we wish to predict which surfaces of the two proteins will come into contact upon binding. A primary challenge for using machine learning for protein interface prediction is the lack of labelled examples, reflecting a more general trend in structural biology: the dearth of task-specific data. As a result, the dominant machine learning approaches in this area have long relied on hand-crafted, high-level features.

While task-specific data is limited, there has been a surge in the availability of protein structures. Furthermore, all this data shares the same underlying feature space $\mathcal{X}_a$: a collection of atoms $a \in \mathbb{A}$ where $\mathbb{A} = \mathbb{P} \times \mathbb{E}$ such that $\mathbb{P} = \mathbb{R}^3$ is the position space and $\mathbb{E} = \{C, N, O, S, ...\}$ is the set of possible atom element types. With this in mind, we set out to address the data-poor task of interface prediction $\mathcal{T}_p$ by designing two key components. First, we define a data-rich task $\mathcal{T}_r$ related to $\mathcal{T}_p$ that would allow us to leverage much larger amounts of protein data. Second, we create an end-to-end classifer that could exploit the unified feature space to transfer its learned features from $\mathcal{T}_r$ to $\mathcal{T}_p$.

Formally, we define an atomic-level task $\mathcal{T}_i$ as follows:

$$\mathcal{T}_i = \{\mathcal{X}_a, \mathcal{Y}_i, P_i(\mathcal{X}_a, \mathcal{Y}_i)\}$$

Where $\mathcal{X}_a$ is the shared atom space described above, $\mathcal{Y}_i$ is the task-specific label space, and $P_i$ is the joint probability distribution over the atom and label spaces. Many tasks other than interface prediction fall under this paradigm, including drug discovery, and protein folding and design.

Returning to the task of interface prediction $\mathcal{T}_p$, we note our sampling of $(x, y) \sim P_p(\mathcal{X}_a, \mathcal{Y}_p)$ is very limited, as the number of cases for which we have experimental structures of both the final complex and for each protein on its own is small. A commonly used comprehensive dataset, Docking

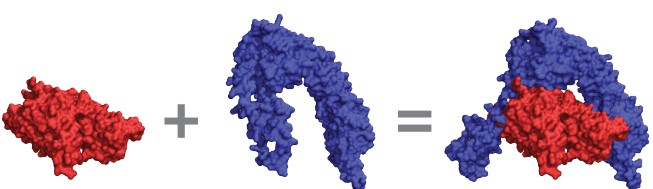

Figure 1: Protein Binding. The BNI1 protein (blue) opens up to bind to Actin (red). While our method is trained only using structures of complexes such as the one at right (sampled from $\mathcal{C}_r$), without any information on how the individual proteins deformed upon binding, we test on pairs of unbound structures such as those at left (sampled from $\mathcal{C}_p$) with minimal loss in performance.

Benchmark 5 (DB5) (Vreven et al., 2015), contains 230 such protein interactions comprising a total of 21,000 neighboring amino acids. The limited size of this set has forced existing methods to rely on hand-engineered structural features (e.g. the depth of an amino acid from the surface of the protein). We have found, however, that we can construct a related, data-rich task $\mathcal{T}_r$: given only the structures of interacting proteins as they bind in their final complex, we wish to predict which surfaces of the two proteins come into contact upon binding. This task is related to $\mathcal{T}_p$ (note that $\mathcal{Y}_p = \mathcal{Y}_r = \{0, 1\}$), but allows us to mine the Protein Data Bank (PDB) (Berman et al., 2000) to obtain over 44,828 binary protein interactions with only the final complex solved experimentally. This leads to over five million neighboring amino acids, an increase of more than two orders of magnitude in the size of the training set. We refer to the former dataset as $\mathcal{C}_p \sim P_p$, and the latter as $\mathcal{C}_r \sim P_r$.

With the much larger dataset $C_r$ in hand, we present SASNet, the first end-to-end learning method applied to interface prediction, and the first method to demonstrate a high degree of transferrability of features learned on $\mathcal{X}_a$. The method is end-to-end because instead of relying on hand-engineered, high-level features, we work directly at the atomic level with the space $\mathbb{A}$. To predict whether an amino acid on the surface of one protein interacts with an amino acid on the surface of another protein, we voxelize the local atomic environments, or "surfacelets," surrounding each of them and then apply a siamese-like three-dimensional convolutional neural network (CNN) to the resulting grids.

We train our end-to-end model on $\mathcal{C}_r$, without accounting for the fact that the proteins deform upon binding. Notably, when tested on $\mathcal{C}_p$, proteins that do deform upon binding, our method outperforms state-of-the-art methods that exploit hand-engineered features and are trained directly on $\mathcal{C}_p$. We also leave open the door to substantially more performance improvements not available to competing models, as we have so far trained on approximately 3% of $\mathcal{C}_r$ (due to computational limitations), whereas standard models are already using all of the $\mathcal{C}_p$ data available to them. Finally, we demonstrate that when trained on $\mathcal{C}_r$, the features learned by competing methods do not transfer well and their performance on $\mathcal{C}_p$ falls dramatically (for the best such method, AUROC is $0.878$ when trained on a subset of $\mathcal{C}_p$, compared to $0.836$ when trained on $\mathcal{C}_r$; further details in Section 5.2).

This is especially exciting as the support of $\mathcal{P}_r$ is a strict subset of the support of $\mathcal{P}_p$ (as in, for any $(x, y) \sim P_r$, if $\mathcal{P}_r(x, y) > 0$ then $\mathcal{P}_p(x, y) > 0$, whereas the converse is not true). This is because protein interfaces must take on a specific configuration upon binding in order to fit together in an energetically favorable manner, whereas as shown in Kuroda & Gray (2016) they have more flexibility when not bound (i.e. the atoms are not as restricted to particular positions, see Figure 1). $\mathcal{C}_r$ only contains proteins in conformations that can already fit together, whereas $\mathcal{C}_p$ also contains protein conformations that require major deformations before being able to fit together. In spite of $\mathcal{C}_r$'s limited coverage of $\mathcal{P}_p$'s more diverse structures, the model's ability to perform well on $\mathcal{C}_p$ indicates the model has not simply memorized the rules governing interaction in $\mathcal{C}_r$ (such as looking for shape complementarity). Instead, it has learned a representation that encodes the flexibility of proteins present in $\mathcal{C}_p$, without being explicitly trained to do so. We argue that the convolutional neural network formulation coupled with raw atomic features has the appropriate form to be able to exploit the high degree of regularity and spatial hierarchy in protein structure, while remaining general enough to learn transferrable features.

## 2 RELATED WORK

Transfer learning has been studied for both unsupervised pre-training and supervised training on unrelated tasks (Bengio, 2011; Long et al., 2015). It is also closely linked to the multi-task learning literature (Liu et al., 2015).

Here, we focus on reviewing the application of such methods to tasks concerning biological structures such as proteins, small drug-like molecules, DNA, and RNA, for which there has been significant interest in applying machine learning methods. The transferrability of these methods to new and unseen tasks has typically not been investigated and has proven unsuccessful in cases where it is considered at all (Ramsundar et al., 2015). Graph-based approaches have been used for deriving properties of small molecules (Kearnes et al., 2016; Duvenaud et al., 2015). Gilmer et al. (2017) used such networks for quantum mechanical calculations. Another common representation for quantum mechanical calculations is based on Behler & Parrinello (2007)'s symmetry functions which use manually determined Gaussian basis functions, as used in (Faber et al., 2017; Smith et al., 2017). Gomes et al. (2017) uses the symmetry functions for protein-ligand binding affinity prediction. Instead of building in invariances, Zhang et al. (2018) canonicalizes the coordinate frame for each atom as well as the ordering of neighbors and trains a fully connected neural network on the result to predict force field potentials and forces. 3D convolutional networks have been used for protein-ligand binding affinity by (Ragoza et al., 2017; Wallach et al., 2015; Jiménez et al., 2017), as well as for protein fold prediction (Derevyanko et al., 2018), and for filling in missing amino acids (Torng & Altman, 2017). Transfer learning has been investigated for the task of protein folding, though this work relies on solely on protein sequence features (Wang et al., 2017). Our work, in contrast to the methods described, represents the first successful application of transfer learning to atomic-level data.

Turning to the problem of interface prediction, methods developed by Fout et al. (2017) and Sanchez-Garcia et al. (2018) have especially high performance (AUROC 0.876 and 0.878, respectively; further details in Table 2). They both apply machine learning techniques (graph convolutions and extreme gradient boosting, respectively) to hand-designed structural features and are trained only on $\mathcal{C}_p$. Other high-performing methods are Jordan et al. (2012), Porollo & Meller (2006), Northey et al. (2018), and Hwang et al. (2016) who also use high-level structural features to predict interfacial residues, but in a non-partner-specific manner – given a single protein, they predict which of its amino acids are likely to form an interface with any other protein. Xue et al. (2015) demonstrates that partner-specific interface predictors yield much higher performance. Our contributions to the problem of interface prediction include both the first use of end-to-end learning and learned structural features that achieve state-of-the-art performance.

Sequence conservation across species is another source of information for addressing the interface prediction problem. The basic idea is that the portions of the protein that are interfacial are typically highly constrained in how they can evolve, as too much variability can interrupt interactions that might be vital to the protein's function. For example, Ahmad & Mizuguchi (2011) uses neural networks trained on such features. Given that all these interfaces are derived from the physics of actual three-dimensional interactions, the relegation of structure to a hidden and unmodeled variable leads to limitations of these approaches. The general consensus in the field is that the performance of purely sequence-based methods is approaching their limit (Esmaielbeiki et al., 2016).

Interface prediction is also of importance to protein–protein docking, the computational task of predicting the three-dimensional structure of a complex from its individual proteins. The space of possible complexes remains vastly under-explored: as of 2017, major databases such as Interactome3D (Mosca et al., 2013) contain a total of approximately 12,000 complexes whose structures have been experimentally determined, while there are estimated to be 650,000 such interactions in humans alone (Stumpf et al., 2008). There are a wide variety of docking methods that have been proposed such as Dominguez et al. (2003), Torchala et al. (2013), Chen et al. (2003), Rezácová et al. (2008), which regularly compete in standardized docking assessments (Janin et al., 2003). Docking software currently achieves low accuracy (Vreven et al., 2015): the lack of robust interface predictors for ranking candidate complexes has been identified as one of the primary issues preventing better performance (Bonvin, 2006).

The primary contribution of this work over the existing literature is demonstrating that end-to-end learning instead of hand-engineering features enables superior transferrability for models trained on data-rich tasks to data-poor tasks involving atomic-level data.

| Dataset | # Binary Complexes | # Amino Acid Interactions |
|---------|--------------------|--------------------------|
| $\mathcal{C}_r$ (PDB) | 44,828 | 5,892,422 |
| $\mathcal{C}_p$ (DB5) | 230 | 21,091 |

Table 1: Dataset Sizes. By training on complexes from $\mathcal{C}_r$ (PDB), as opposed to restricting ourselves to complexes with unbound data available such as those from $\mathcal{C}_p$ (DB5), we can access over two orders of magnitude more training data than would otherwise be available.

## 3 DATASET

We construct two separate datasets for testing and training our method. The first, $\mathcal{C}_p$, is our gold standard data-limited dataset which we use for testing performance. It comprises the 230 protein complexes in the Docking Benchmark 5 (DB5) dataset (Vreven et al., 2015). Interfacial amino acids (i.e., the labels) are defined based on the final bound complex, but the 3D structures used as input to the model are derived from the individual unbound proteins. The data distribution therefore closely matches that which we would see when predicting interfaces on new examples, which will be provided in their unbound states. Additionally, the range of difficulty and of interaction types in this dataset (e.g. enzyme-inhibitor, antibody-antigen) provides us with good coverage of typical test cases we might see in the wild. Previous state-of-the-art methods, which can only leverage $C_p$, train on on the first 140 complexes of $\mathcal{C}_p$, validate on the next 35 complexes, and use the final 55 as a test set (Fout et al., 2017; Sanchez-Garcia et al., 2018). We refer to these as $\mathcal{C}_p^{train}$, $\mathcal{C}_p^{val}$, and $\mathcal{C}_p^{test}$, respectively. $\mathcal{C}_p^{test}$ represents the most recently released complexes in DB5, and its use as a testing set allows for an estimation of these method's performance on unreleased data.

To construct $\mathcal{C}_r$, our more data-rich dataset, we start by mining the PDB for pairs of interacting protein subunits (Figure 2A). For this dataset, both the input structures to the model and the labels are derived from the final, bound complex. The PDB contains data of varying quality, so we only include complexes that meet the following criteria: $\geq 500\text{Å}^2$ buried surface area, solved using X-ray crystallography or cryo-EM at better than 3.5Å resolution, only contains protein chains longer than 50 amino acids, and is the first model in a structure. Furthermore, DB5 is initially derived from the PDB, so we use sequence-based pruning to ensure that there is no complex-level cross-contamination between our train and test sets. Specifically, we exclude any complex that has any individual protein with over 30% sequence identity when aligned to any protein in $\mathcal{C}_p$. This is a commonly-used sequence identity threshold (Yang et al., 2013; Jordan et al., 2012). The initial processing as well as the DB5 exclusion yields a dataset of 44,828 binary complexes. Note that competing methods do not employ such pruning on their training set — a potential source of bias. Alternate exclusion criteria do not significantly impact model performance ($0.890 \pm 0.011$ for the dataset as described above; $0.887 \pm 0.007$ using a 20% sequence identity cutoff; $0.883 \pm 0.007$ after removal of complexes sharing any domain-domain interaction with the test set using 3did (Mosca et al., 2014)).

For both of these datasets, once these binary protein complexes are generated, we identify all interacting pairs of amino acids. A pair of amino acids — one from each protein — is determined to be interacting if any of their non-hydrogen atoms (hydrogen atoms are typically not observed in experimental structures) are within 6Å of one another (Figure 2B). This 6Å threshold is commonly used (Fout et al., 2017; Sanchez-Garcia et al., 2018). We consider each of these pairs as a positive example of interacting surfaces, leading to a total of over 5 million pairs of positives for the PDB dataset (Figure 2C, see Table 1 for exact counts). For the negatives, we select random pairs of non-interacting amino acids spanning the same protein complexes, ensuring a fixed ratio of positives to negatives for each complex (Figure 2D, ratio found via hyperparameter search, see section 4).

As noted previously, the structures in PDB Dataset $\mathcal{C}_r$ are already in their bound state. Typical methods to solve the interface prediction problem are trained on much smaller datasets containing unbound proteins (with labels derived from the final bound complex), such as our DB5 dataset $\mathcal{C}_p$. A key point of this work is that we leverage the much larger $\mathcal{C}_r$ to solve the problem of protein interface prediction on $\mathcal{C}_p$. The transferrability between these two problems is not obvious. For example, $\mathcal{C}_r$ has a much higher degree of shape complementarity than $\mathcal{C}_p$, as the former exclusively comprises pairs of proteins that are all in the correct configuration to interact with each other.

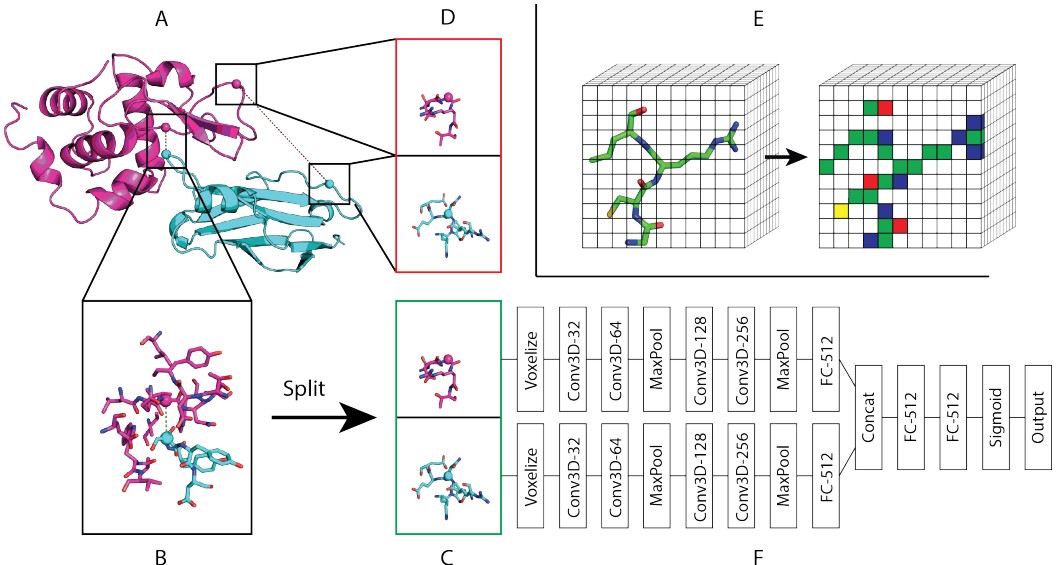

Figure 2: Protein Interface Prediction via SASNet. We predict which parts of two proteins have the potential to interact by constructing a binary classifier. To extract training examples for the problem, we start with a pair of proteins in complex sampled from $\mathcal{C}_r$ (A, proteins shown in cartoon form), and from there extract all pairs of interacting amino acids (B, atoms shown in stick form). We then split these pairs to obtain our positives (C), as well as sampling random non-interacting pairs from the complex for our negatives (D). These pairs are then individually voxelized into 4D grids, the last dimension being the one-hot encoding of the atom's element type (E, atom channel shown as color). These pairs of voxelized representations are then fed through a 3D siamese-like CNN (F).

## 4 METHOD

Due to the hierarchical and regular structure of proteins, as well as the wealth of data available for the protein interface prediction, we selected a three-dimensional convolutional neural network as SASNet's underlying model (Figure 2F). We first focus on how to represent our pairs of amino acids in order to provide them to our network. For each amino acid, we encode its surrounding atomic neighborhood of $n$ atoms as $\mathbb{A}^n$ — a region of 3D space centered around its alpha-carbon which we call a "surfacelet." This encodes all structural data local to this central alpha-carbon that is provided in a given PDB structure.

To create a dense, three-dimensional, and fixed-size representation of the input, we choose to voxelize the space (Figure 2D). For each surfacelet, we lay down a grid centered on the alpha carbon of the amino acid, and record in each voxel the presence or absence of a given atom. To ensure that at most one atom can occur in each voxel, while also keeping the input representation from getting too large, we chose a voxel resolution of 1Å. A fourth dimension is used to encode the element type of the atom, using 4 channels for carbon, oxygen, nitrogen, and sulfur, the most commonly found atoms in protein structure. In order to build in a notion of rotational invariance, each training example is randomly rotated, every time it is seen, across the 3 axes of rotation. At test time, we perform 20 random rotations for each example and average the predictions.

Choices for the following architecture were validated using manual hyperparameter search as described below. We feed the voxelized surfacelets to multiple layers of 3D convolution (Conv3D) followed by batch normalization (BN) and rectified linear units (ReLU), and optionally layers of 3D max pooling (MaxPool). We then apply several fully-connected (FC) layers followed by more BNs and ReLUs. As we are working with pairs of surfacelets, we use a siamese-like networks where we employ two such networks with tied weights to build a latent representation of the two surfacelets, and then concatenate the results. An important difference compared to classical siamese approaches, as introduced by Bromley et al. (1993), arises from the nature of the task we are predicting. Unlike a classical siamese network, we are not attempting to compute a similarity between two objects. This

| Method | CAUROC |
|---|---|
| NGF (Duvenaud et al., 2015) | 0.843 (0.851 +/- 0.010) |
| DTNN (Schütt et al., 2017) | 0.861 (0.861 +/- 0.004) |
| Node+Edge Average (Fout et al., 2017) | 0.844 (0.850 +/- 0.004) |
| Order Dependent (Fout et al., 2017) | 0.857 (0.864 +/- 0.006) |
| Node Average (Fout et al., 2017) | 0.876 (0.877 +/- 0.005) |
| BIPSPI (Sanchez-Garcia et al., 2018) | 0.878 (0.878 +/- 0.003) |
| **SASNet** | **0.892 (0.885 +/- 0.009)** |

Table 2: $\mathcal{C}_p^{test}$ CAUROC performance. For each method we report best replicate (as selected by $\mathcal{C}_p^{val}$ performance for competitors, and by $\mathcal{C}_r^{val}$ for SASNet) as well as mean and standard deviation across replicates. While competing methods have used all available training data, due to computational limitations SASNet is trained on less than 3% of $\mathcal{C}_r$, hinting at the possibility of substantial performance improvements.

can be shown by considering the nature of protein interactions: a positively charged protein surface is likely to interact with a negatively charged counter-part, even though the two could be considered very dissimilar. Instead of minimizing Euclidean distance between the two latent representation as would be done in a classical siamese network, we append a series of fully connected layers on the concatenation of the two latent representations and optimize the binary cross entropy loss with respect to the original training labels.

To determine the optimal model, we ran a large set of manual hyperparameter searches on a limited subset of the full PDB dataset, created based on selection criteria from (Kirys et al., 2015). We vary the number of filters, number of convolutional layers, number of dense layers, the ratio of class imbalance, grid size, and use of maxpooling, batchnorm, and dropout, and selected our models based on average performance across three or more replicates. Each replicate is trained on a randomized subset of a randomly selected training and validation set, referred to as $C_r^{train}$ and $C_r^{val}$ and consisting of 623 and 77 complexes, respectively. Surprisingly, most of the parameters had little effect on the overall validation performance, with the exception of the positive effect of increasing grid size.

Our model with the best validation performance involved featurizing a grid of edge length 35Å (thus starting at a cube size of 35x35x35), and then applying 4 layers of convolution (with filter sizes 32, 64, 128, and 256) and 2 layers of max pooling, as shown in Figure 2F. A fully-connected layer with 512 parameters lays at the top of each tower, and the outputs of both towers are concatenated and passed through two more fully connected layers with 512 parameters each, leading to the final prediction. The number of filters used in each convolutional layer is doubled every time to allow for an increase of the specificity of the filters as the spatial resolution decreases. We use the RMSProp optimizer with a learning rate of 0.0001. The positive-negative class imbalance was set to 1:1. The overall network is designed such that the grid feeding into the first dense layer is not of too great a size to cause memory issues yet not too small to lose all spatial information. All models are trained across 4 Titan X GPUs using data-level parallelism.

## 5 EXPERIMENTS

The combination of a dense featurization, large datasets, and a model that exploits the inherent structure of proteins allows us to outperform state-of-the-art methods on standardized and well-curated benchmarks, while making almost no assumptions with respect to the problem of protein interface prediction. Furthermore, we demonstrate the superior transferrability of the model's learned features by training competing methods on $\mathcal{C}_r$ and testing on $\mathcal{C}_p$. Finally, we observe the model's scalability, noting that there is potential for further performance improvements via scaling to a larger fraction of the training dataset. All reported models were run across a minimum of 3 replicates.

| Method | CAUROC |
|---|---|
| Node Average (Fout et al., 2017) | 0.712 (0.714 +/- 0.022) |
| BIPSPI (Sanchez-Garcia et al., 2018) | 0.836 (0.836 +/- 0.001) |
| **SASNet** | **0.892 (0.885 +/- 0.009)** |

Table 3: $\mathcal{C}_p^{test}$ CAUROC performance for leading methods trained on $\mathcal{T}_r$ task. Competing methods with hand-engineered features experience a dramatic drop in performance as compared to their performance in Table 2. The features learned by SASNet exhibit a higher degree of transferrability.

## 5.1 COMPARISON TO EXISTING INTERFACE PREDICTION METHODS

We start by evaluating the effectiveness of our structural features by comparing to top existing methods applied to $\mathcal{T}_p$, as shown in Table 2. Some of these methods were pulled from the comparison in Fout et al. (2017) and include Deep Tensor Neural Networks (DTNN) from Schütt et al. (2017), and Neural Graph Fingerprints (NGF) from Duvenaud et al. (2015). Another state-of-the-art feature-engineering tree ensemble method is BIPSPI (Sanchez-Garcia et al., 2018). Our goal is to maximize performance of structural features, and so in order to isolate the effectiveness of the structural representations, we remove sequence features from the compared models and re-run their training procedures. Results with sequence added in are reported in Appendix A.

For each model, we select from available hyperparameters by evaluating $\mathcal{C}_p^{val}$ ($\mathcal{C}_r^{val}$ for SASNet) performance across replicates. At test time we evaluate on $\mathcal{C}_p^{test}$, splitting the predictions by complex and computing the Area Under the Receiver Operating Characteristic (AUROC) for each one. We then calculate the median of those AUROCs. We refer to this as the median per-Complex AUROC (CAUROC). As our final performance metric we report the mean and standard deviation of CAUROC across all replicates, as well as the CAUROC of the replicate with the best validation performance. Our models demonstrate superior performance to all other methods without the use of any hand-engineered features, and without using $\mathcal{C}_p$ for any part of the training or validation pipeline.

## 5.2 TRANSFERRABILITY

A natural question to ask is whether SASNet's performance gains are simply due to the use of the larger $\mathcal{C}_r$ data set for training. If $P_r$ and $P_p$ were overly similar distributions, then it would be relatively straightforward to leverage the larger size of $\mathcal{C}_r$ to improve performance. As $KL(P_r \| P_p) \to 0$ we would observe that $\mathbb{E}_{(x,y) \sim P_p}[L(x, y; \theta_r)] \geq \mathbb{E}_{(x,y) \sim P_p}[L(x, y; \theta_p)]$ given the loss $L$ of classifiers with parameters $\theta_p$ and $\theta_r$ derived from training on $\mathcal{C}_p$ and the larger set $\mathcal{C}_r$, respectively. We show that this is not the case by investigating the contrapositive — taking a classifier initially trained on $\mathcal{C}_p$ and instead training it on $\mathcal{C}_r$, and evaluating the change in performance.

We run this procedure on both our own model (effectively our existing training pipeline) and on the two competing methods with the highest performing structural features, BIPSPI (Sanchez-Garcia et al., 2018) and Node Average (Fout et al., 2017). As shown in Table 3 as compared to Table 2, instead of staying even or increasing, the performance of competing methods degrades dramatically when trained on $\mathcal{C}_r$ as opposed to $\mathcal{C}_p$, indicating that $P_r$ and $P_p$ are not similar. This likely arises because the hand-engineered features used by previous methods assume that $\mathcal{C}_r$ is in the unbound form, like $\mathcal{C}_p$. Our method, on the other hand, is robust to the use of $\mathcal{C}_r$ for training, allowing us to use the larger training dataset successfully. This comparison is a clear demonstration of why a model making minimal feature assumptions can be advantageous for atomic data. A caveat to note is that while the detailed nature of the learned features permits this high degree of transferrability, classical feature engineering methods may be a better fit when structural information is less detailed.

## 5.3 HYPERPARAMETER EFFECTS

Given the expense of running 3D convolutions, our best models are trained on a fraction of the full dataset $\mathcal{C}_r$ available to us. We are additionally limited by the size and resolution of the grids due to the cubic relationship between edge size and the total number of voxels. Finally we are restricted in the number of rotation augmentations performed per data point. As these problems are

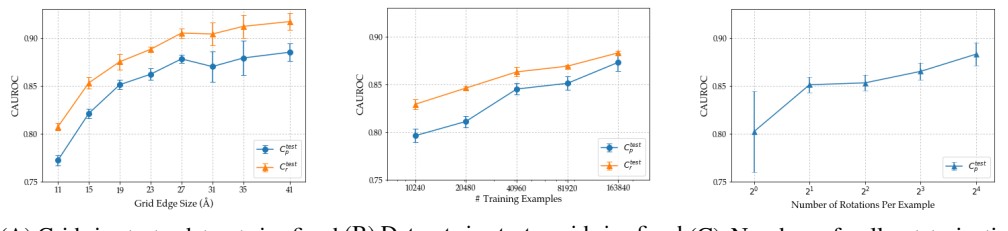

(A) Grid size tests, dataset size fixed to 81920.
(B) Dataset size tests, grid size fixed to 23Å.
(C) Number of rolls at train time tests, grid size fixed to 23Å.

Figure 3: Model Scaling. Mean CAUROC is reported, with standard deviation marked as error bars.

surmountable through engineering effort, we are interested in assessing the benefits of scaling up. We run 5 replicates per condition and plot average and standard deviation of CAUROC across replicates.

Figure 3A shows the results of the grid size scaling tests. We notice consistent performance improvements up to a grid edge size of 27Å, with performance increases becoming noisier and mostly tapering off afterwards. We do note that extra signal is still being gained at very large sizes such as 41Å, implying a long-range contribution of forces from atoms distant to the central amino acid. In Figure 3B, we see that the dataset size tests yield consistently increasing performance, reflecting the high degree of scalability of our model, and implying that further performance gains could be obtained with larger dataset sizes. Finally, we show in Figure 3C that decreasing the number of rotational augmentations per example at train time degrades performance.

## 6 Conclusion

In this work we introduced the first end-to-end learning framework to predict protein interfaces in conjunction with the first successful application of transfer learning to atomic-level data. We surpass current state-of-the-art results on the general interface prediction problem $\mathcal{T}_p$ while only training on the task $\mathcal{T}_r$ of predicting interfaces for proteins already in their bound configurations, without using any expert feature identification. This is particularly intriguing as proteins are flexible structures, that can deform at multiple scales, and the task $\mathcal{T}_r$ gives us only a small subset of the possible shapes proteins can adopt (since they must be in a specific shape to bind). The high performance on $\mathcal{T}_p$ indicates our model is able to generalize to configurations beyond those provided in $\mathcal{T}_r$, showing it has learned a notion of protein flexibility without being trained to do so.

The small number of assumptions made and transferrability of the learned features are also of interest, as we can envision solving many data-poor problems involving $\mathcal{X}_a$ (such as protein design and drug discovery) through training on larger, tangentially related datasets. Much like an enhanced version of pre-training on ImageNet for computer vision tasks, we could employ pretrained models in structural biology that have already learned to encode many of the patterns present in biomolecular structures, allowing us to solve new tasks with minimal to no re-adaptation necessary.

One hypothesis as to why SASNet's CNNs are able to transfer so well for these tasks is that proteins are highly spatially hierarchical and regular in nature, as well as being governed by the same underlying laws of physics, making them a good fit for the stacked convolutional framework. Though these properties are well understood at the lowest levels (only 22 amino acids are genetically encoded, each having a fixed atomic composition), the definitions become less precise as we move up the hierarchy. Amino acids often form secondary structure elements such as alpha-helices and beta-sheets. At a higher level, parts of the protein can form into independent and stable pieces of 3D structure known as protein domains. Finally, whole proteins can be built out of these domains. Many motifs are shared between proteins at all levels of this hierarchy. Current schemes for classifying protein structure often rely on manually curated hierarchies (Andreeva et al., 2014) that are not able to cleanly capture all possible variations. Thus, CNNs may be able not only to capture the known relationships between structural elements at different scales, but also to derive new relations that have not been fully characterized. Further investigation of the learned filters could yield insight into the nature of these higher-level structural patterns, allowing for a better understanding of protein structure.

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

## APPENDIX A  SEQUENCE-ADDED RESULTS

As shown in Table 4, a combination of SASNet's final learned layer and Fout et al. (2017)'s sequence conservation features allows SASNet to surpass their performance. The exact architecture is a linear combination of the aforementioned features, fed through a sigmoid activation function. Only this final layer is updated during re-training, with the rest of the SASNet network being held fixed. Due to the computational expense of deriving sequence conservation features we restrict our training and validation to $C_p^{train}$ and $C_p^{val}$, while still testing on $C_p^{test}$. While Sanchez-Garcia et al. (2018) achieve higher overall performance, they also make use of additional sequence correlation features not used by Fout et al. (2017) (and by extension SASNet).

| Method | CAUROC |
|---|---|
| NGF (Duvenaud et al., 2015) | 0.869 (0.875 +/- 0.018) |
| DTNN (Schütt et al., 2017) | 0.869 (0.870 +/- 0.003) |
| Node+Edge Average (Fout et al., 2017) | 0.895 (0.899 +/- 0.005) |
| Order Dependent (Fout et al., 2017) | 0.896 (0.894 +/- 0.004) |
| Node Average (Fout et al., 2017) | 0.887 (0.888 +/- 0.004) |
| **BIPSPI (Sanchez-Garcia et al., 2018)** | **0.942 (single replicate)** |
| SASNet | 0.921 (0.914 +/- 0.009) |

Table 4: $C_p^{test}$ CAUROC performance for methods trained on structure and sequence. SASNet adds in sequence by training a linear model, followed by a sigmoid activation, on the concatenation of the sequence conservation features used in Fout et al. (2017) and the final layer of the best learned structural SASNet model.

