# OpenReview forum: "Transferrable End-to-End Learning for Protein Interface Prediction"
_ICLR.cc/2019/Conference_

### Official Review · AnonReviewer1 · 2018-11-03
**good idea but unclear model**

**Rating:** 5
**Confidence:** 4

**Review:**

This manuscript applies transfer learning for protein surface prediction. The problem is important and  the idea is novel and interesting. However, the  transfer learning model is unclear.
Pros:  interesting and novel idea
Cons:  unclear transfer learning model, insufficient experiments.

Detail: section 4 describes the transfer learning model used in the work, but the description is unclear. It is unknown the used model is a new model or existing model. Besides, in the experiments, the proposed method is not compared to other transfer learning methods.  Thus, the evidence of the experiments is not enough.

---

> ### Author Response · Authors · 2018-11-14
> **Response to Reviewer 1**
>
> Response: We thank the reviewer for their comments.  We would like to clarify one of the central points of this paper, as the cons presented are built upon a misunderstanding of this point.  We are not proposing a new transfer learning model -- we are demonstrating the transferrability of the atomic features we have learned.  We train our structural features on C_r and show that with no re-training they can achieve state-of-the-art results of C_p.  Applying a classical transfer learning algorithm might improve performance even further, as then we could fine-tune results on C_p.  Though this is an interesting direction, it is outside the scope of the work we present here, which concerns itself with the learned representations themselves.  Thus, instead of comparing transfer learning methods, we evaluate the transferrability of both our own structural features as well as those of competitors.

---

### Official Review · AnonReviewer2 · 2018-11-03
**Decent application paper and setup for siamese networks**

**Rating:** 5
**Confidence:** 3

**Review:**

Summary:
This paper uses siamese networks to define a discriminative function for predicting protein-protein interaction interfaces. They show improvements in predictive performance over some other recent deep learning methods.
The work is more suitable for a bioinformatics audience though, as the bigger contribution is on the particular application, rather than the model / method itself.

Novelty:
The main contribution of this paper is the representation of the protein interaction data in the input layer of the CNN

Clarity:
- The paper is well written, with ample background into the problem.

Significance:
- Their method improves over prior deep learning approaches to this problem. However, the results are a bit misleading in their reporting of the std error. They should try different train/test splits and report the performance.
- This is an interesting application paper and would be of interest to computational biologists and potentially some other members of the ICLR community
- Protein conformation information is not required by their method

Comments:
- The authors should include citations and motivation for some of their choices (what sequence identity is used, what cut-offs are used etc)

-  The authors should compare to at least some popular previous approaches that use a feature engineering based methodology such as - IntPred

- The authors use a balanced ratio of positive and negative examples. The true distribution of interacting residues is not balanced -- there are several orders of magnitude more non-interacting residues than interacting ones. Can they show performance at various ratios of positive:negative examples? In case there is a consistent improvement over prior methods, then this would be a clear winner

---

> ### Author Response · Authors · 2018-11-14
> **Response to Reviewer 2**
>
> We thank the reviewer for their comments.  We address their comments individually below.
>
> > The work is more suitable for a bioinformatics audience though, as the bigger contribution is on the particular application, rather than the model / method itself.  The main contribution of this paper is the representation of the protein interaction data in the input layer of the CNN
>
> Response: The main contribution and novelty of this paper is the demonstration of the transferrability and power of the learned representation, and thus is a good fit for ICLR.  We use the application of protein interface prediction as a test case for this, but applications can range widely from drug discovery, to RNA folding, to small molecule quantum mechanical calculations.   As many of these tasks are very data-poor, this demonstrated transferrability opens up novel avenues through which these problems can be tackled.
>
> > - Their method improves over prior deep learning approaches to this problem. However, the results are a bit misleading in their reporting of the std error. They should try different train/test splits and report the performance.
>
> Response: We do use different subsets of the train set for different replicates.  However, the train and test sets cannot be mixed as they come from different data distributions (P_r and P_p) and we are trying to show we can transfer with no retraining from P_r to P_p.  Thus our reported metrics are correct and justified for this problem, though we have clarified the exact nature of the replicates in the text to ensure this is not misleading.
>
> > - The authors should include citations and motivation for some of their choices (what sequence identity is used, what cut-offs are used etc)
>
> Response: We do provide citations for these choices.  See the second and third paragraphs of section 3 on page 4 for motivation/citations for sequence identity and cut-offs used, respectively.
>
> > -  The authors should compare to at least some popular previous approaches that use a feature engineering based methodology such as - IntPred
>
> Response: Fout et al. and Sanchez-Garcia et al. are feature engineering approaches -- they both use high-level features as inputs to their models (not atomic coordinates).  Sanchez-Garcia et al. use a tree ensemble model that has no end-to-end learning aspects at all.  Another popular pure feature engineering approach is PAIRPred (Minhas et al., Protein 2014), which uses an SVM trained on high-level features.  However, we do not compare to them as their performance on C_p^{test} (0.863)  was already superseded in Fout et al.’s work. IntPred [Northey et al., Bioinformatics 2017] addresses the binding site prediction problem (given one protein, which residues can be interfacial with any other protein), which is different than the problem we present.
>
> > - The authors use a balanced ratio of positive and negative examples. The true distribution of interacting residues is not balanced -- there are several orders of magnitude more non-interacting residues than interacting ones. Can they show performance at various ratios of positive:negative examples? In case there is a consistent improvement over prior methods, then this would be a clear winner
>
> Response: We can demonstrate consistent performance at different ratios of positive:negative examples.  Running tests on C_p^{test} at 1:3, 1:5, and 1:10 ratios demonstrate no significant impact on performance (0.889 [0.882 +/- 0.012], 0.889 [0.882 +/- 0.011], and 0.895 [0.886 +/- 0.015], respectively).  The AUROC metric we use is insensitive to class imbalance, and thus is a good measure to use when evaluating on datasets with varying amounts of imbalance.

---

### Official Review · AnonReviewer3 · 2018-11-04
**Nice writing. Lack of significant contribution. Insufficient experimental evidence.**

**Rating:** 5
**Confidence:** 3

**Review:**

For the task of predicting interaction contact among atoms of protein complex consisting of two interacting proteins, the authors propose to train a Siamese convolutional neural network, noted as SASNet, and to use the contact map of two binding proteins’ native structure.
The authors claim that the proposed method outperforms methods that use hand crafted features; also the authors claim that the proposed method has better transferability.

My overall concern is that the experiment result doesn’t really fully support the claim in the two aspects: 1) the SASNet takes the enriched dataset as input to the neural net but it also uses the complex (validation set) to train the optimal parameters, so strictly it doesn’t really fit in the “transfer” learning scenario. Also, the compared methods don’t really use the validation set from the complex data for training at all. Thus the experiment comparison is not really fair. 2) The experiment results include standard errors for different replicates where such replicates correspond to different training random seeds (or different samples from the enriched set?), however, it doesn’t include any significance of the sampling. Specifically, the testing dataset is fixed. A more rigorous setting is to, for N runs, each run splitting the validation and testing set differently.

Since this paper is an application paper, rather than a theoretical paper that bears theoretical findings, I would expect much more thorough experimental setup and analysis. Currently it is still missing discussion such as, when SASNet would perform better and when it would perform worse, what it is that the state of the art features can’t capture while SASNet can. Moreover, it is the prediction performance that matters to such task, but the authors remove the non-structure features from the compared methods. Results and discussion about how the previous methods with full features perform compared to SASNet, and also how we can include those features into SASNet should complete the paper.

Overall the paper is well written, and I do think the paper could be much stronger the issues above are addressed.


Some minor issues:
1)	on page 4, Section 3, the first paragraph, shouldn’t “C_p^{val} of 55” be “C_p^{test} of 55”?

2)	It is not clear what the “replicates” refer to in the experiments.

3)	Some discussion on why the “SASNet ensemble” would yield better performance would be good; could it be overfitting?

---

> ### Author Response · Authors · 2018-11-14
> **Response to Reviewer 3 [1/2]**
>
> We thank the reviewer for their comments.  We address their comments individually below.
>
> > My overall concern is that the experiment result doesn’t really fully support the claim in the two aspects: 1) the SASNet takes the enriched dataset as input to the neural net but it also uses the complex (validation set) to train the optimal parameters, so strictly it doesn’t really fit in the “transfer” learning scenario.
>
> Response: Our work is indeed not classical transfer learning -- it is in fact an even stricter variant.  We do not re-train the parameters of the neural network at all using C_p, which is typically done as a “fine-tuning” step in the transfer learning scenario. So while we do use C_p^{val} for model selection (i.e., hyperparameter tuning), this is still much less use of the data-poor dataset than in the common transfer learning setting of actually fine-tuning the parameters of the neural network using a subset of the data from the data-poor dataset.
>
> The use of C_p^{val} for hyperparameter tuning was incidental and not a central point of our paper.  To really make this clear, we have updated the paper to demonstrate that even if we do not use C_p^{val} for model selection, and instead select from the same class of models we previously generated by using a randomly selected held-out set C_r^{val}, we still obtain state-of-the-art performance (0.892 [0.885 +/- 0.009]).  In this formulation, C_p is not used at all by our method until test time.
>
> > Also, the compared methods don’t really use the validation set from the complex data for training at all. Thus the experiment comparison is not really fair.
>
> The competing models do make use of validation set C_p^{val} from the complex data to select amongst the most important hyperparameters of their model -- which is equivalent to what we did in our initial formulation, and favors the competing methods compared to if we use C_r^{val} for hyperparameter search instead.
>
> > 2) The experiment results include standard errors for different replicates where such replicates correspond to different training random seeds (or different samples from the enriched set?), however, it doesn’t include any significance of the sampling. Specifically, the testing dataset is fixed. A more rigorous setting is to, for N runs, each run splitting the validation and testing set differently.
>
> Response: The replicates correspond to different training and validation samples of the enriched set -- we have clarified this in the paper.  While it is true that the hyperparameter validation set was initially fixed, the switch to use C_r^{val} as above resolves this. The testing data C_p^{test} is that which has been used in the prior works we compare to (Fout et al. 2017; Sanchez-Garcia et al. 2018).  Furthermore, use of this subset for performance evaluation is justified as as C_p^{test} corresponds to latest released structures in C_p, leading to a more accurate assessment of how such methods would perform on unreleased structures (as they do no sequence identity pruning).  Thus our experimental set up is rigorous and justified.
>
> > Since this paper is an application paper, rather than a theoretical paper that bears theoretical findings, I would expect much more thorough experimental setup and analysis. Currently it is still missing discussion such as, when SASNet would perform better and when it would perform worse, what it is that the state of the art features can’t capture while SASNet can.
>
> Response: As we discuss above, we believe our experimental setup and analysis is sufficient to demonstrate that our atomic representation transfers much better across atomic tasks.  We have also added to our discussion, making clear that our method represents a significant advantage over competing methods when detailed atomic information is available.  Competitors rely on amino acid-level features that fail to capture specific atomic positions but can be better when the structural is less detailed or accurate.

---

> > ### Author Response · Authors · 2018-11-14
> > **Response to Reviewer 3 [2/2]**
> >
> > > Moreover, it is the prediction performance that matters to such task, but the authors remove the non-structure features from the compared methods. Results and discussion about how the previous methods with full features perform compared to SASNet, and also how we can include those features into SASNet should complete the paper.
> >
> > Response: As our paper is primarily about the power and transferrability of our structural features for atomic tasks, we believe a detailed investigation of non-structural features is mostly outside of the scope of this work.  To show that we can easily include these features, we have included in our appendix some results including non-structural features.  When adding in the sequence features used by Fout et al. via a simple linear model combining our final hidden layer and the additional sequence features, we are able to achieve a superior performance of 0.921 (0.914 +/- 0.009) versus their performance of 0.896 (0.894 +/- 0.004).  While BIPSPI (Sanchez-Garcia et al. 2018) does achieve the best combined performance at 0.942, they also use additional sequence correlation features (note their structure-only performance is comparable to that of Fout et al).
> >
> > > Some discussion on why the “SASNet ensemble” would yield better performance would be good; could it be overfitting?
> >
> > Response: We have removed the SASNet ensemble from the paper, as it was based on C_p^{val} and confuses the point we are making about minimally relying on C_p for training and validation.  We could definitely investigate further why this mild ensembling yields a small performance increase, but we see this as tangential to the overarching points of the paper.

---

### Meta-Review · Area_Chair1 · 2018-12-14
**No strong reviewer support**

**Confidence:** 4
**Recommendation:** Reject

**Metareview:**

Two out of three reviews for this paper were provided in detail, but all three reviewers agreed unanimously that this paper is below the acceptance bar for ICLR. The reviewers admired the clarity of writing, and appreciated the importance of the application, but none recommended the paper for acceptance due largely to concerns on the experimental setup.